# Salivary Profile Analysis Based on Oral Cancer Risk Habits: An Observational Cross-Sectional Study

**DOI:** 10.3390/biomedicines12081748

**Published:** 2024-08-02

**Authors:** Rahmi Amtha, Indrayadi Gunardi, Armelia Sari Widyarman, Tiffany Herwanto, Firstine Kelsi Hartanto, Vui King Vincent-Chong

**Affiliations:** 1Oral Medicine Department, Faculty of Dentistry, Universitas Trisakti, Jakarta 11440, Indonesia; rahmi.amtha@trisakti.ac.id (R.A.); firstine@trisakti.ac.id (F.K.H.); 2Oral Microbiology Department, Faculty of Dentistry, Universitas Trisakti, Jakarta 11440, Indonesia; armeliasari@trisakti.ac.id; 3Faculty of Dentistry, Universitas Trisakti, Jakarta 11440, Indonesia; tiffanyherwanto140@gmail.com; 4Department of Oral Oncology, Roswell Park Comprehensive Cancer Center, Buffalo, NY 14263, USA

**Keywords:** risk factor, saliva, oral cancer

## Abstract

Background: In Indonesia, cultural practices such as betel quid chewing, smoking, and alcohol consumption are prevalent. These practices are known risk factors for oral cancer and may influence the salivary profile, which is essential for maintaining oral health. Purpose: To compare the salivary profiles of individuals with and without risk factors for oral cancer. Methods: The study included 49 individuals identified as having risk factors for oral cancer. Unstimulated saliva samples were collected. Various parameters were measured, including salivary pH, flow rate (FR), thickness, color, turbidity, and the levels of IL-1β and IL-8. Data were analyzed using Chi-square and *t*-tests. Results: A significant difference was found in salivary IL-1β levels between the two groups (*p* = 0.009), with higher levels observed in individuals with oral cancer risk factors. Notably, the salivary IL-1β concentrations showed significant differences between the smoking group (*p* = 0.021; OR = 2.94) and the alcohol-drinking group (*p* = 0.007; OR = 4.96) compared to the control group. However, no significant differences were observed between the groups in terms of salivary viscosity, color, turbidity, flow rate, acidity, or IL-8 levels (*p* > 0.05). Conclusion: Individuals with risk factors for oral cancer exhibit distinct salivary IL-1β profiles compared to those without such risk factors, particularly those who practice alcohol drinking.

## 1. Introduction

The World Health Organization (WHO) estimates that around 58% of global oral and oropharyngeal cancer cases occur in South and Southeast Asia, including Indonesia [1]. Various risk factors contribute to the development of oral cancer, including tobacco smoking, alcohol consumption, chronic inflammation, human papillomavirus (HPV), genetic predisposition, and dietary habits [2,3].

Each province in Indonesia has its distinct traditions and cultures. The practice of chewing betel quid, a tradition passed down through generations, is prevalent. In 2004, the International Agency for Research on Cancer (IARC) classified betel quid chewing, with or without tobacco, as carcinogenic to humans [4]. The consumption of alcoholic beverages, particularly a traditional alcohol known as “moke” or “sopi” derived from palm tree sap with a high ethanol content, is another prevalent habit among the Flores population [5]. According to the 2018 Indonesian National Health Survey, smoking prevalence in Indonesia is significant, with East Nusa Tenggara having notably high rates of 19% and non-routine smokers at 7.3%. It is also the second-highest province with a proportion of the consumption of alcoholic beverages in the past month of 15.6%. Tobacco exposure, a component of smoking, is a significant risk factor for oral cancer [6]. 

Saliva plays a crucial role in maintaining oral health through various functions and properties. It acts as a lubricant, aiding in chewing, swallowing, and speaking, while forming a protective barrier against mechanical damage and pathogens [7]. Saliva, as a bodily fluid, reflects various physiological and pathological conditions. It is easily accessible, cost-effective, and non-invasive, making it an ideal medium for diagnostic purposes [8,9]. Saliva has a buffering capacity that neutralizes acids produced by bacteria, preventing tooth enamel demineralization and promoting remineralization. Its antimicrobial agents, such as lysozyme and lactoferrin, control the growth of oral microbiota and prevent infections [10]. It also acts as a solvent for taste substances, facilitating interaction with taste receptors and maintaining taste buds [11]. Saliva is primarily composed of water (99%) along with electrolytes, mucus, antibacterial compounds, and enzymes. Its viscosity, influenced by mucin glycoproteins, is essential for its lubricating function [12]. Salivary flow rate varies with factors like hydration and diet, with a normal unstimulated flow at 0.3–0.4 mL/min and stimulated flow at 1–2 mL/min [13]. The pH of saliva ranges from 6.2 to 7.6, maintained by bicarbonate, phosphate, and protein systems, contributing to its buffering capacity [14]. In the presence of a tumor, the pH of saliva becomes acidic. This change is due to the anaerobic metabolism of glucose under the hypoxic conditions created by the tumor. The acidic environment provides a supportive setting for a tumor cell to survive and proliferate [15]. Saliva contains proteins, including cytokines, which have been identified as potential biomarkers for various diseases, including oral cancer. IL-8 and IL-1β, cytokines implicated in processes such as replication, angiogenesis, and tumor development, are particularly relevant in oral cancer diagnosis [15,16,17]. Several studies have indicated that salivary cytokines may serve as diagnostic biomarkers for various diseases, including oral cancer. Pro-inflammatory cytokines such as IL-1β, IL-6, IL-8, INF-γ, and TNF-α are known to enhance cell growth, disrupt tumor suppression mechanisms, and impair host immunity, leading to cancer progression. Hence, over 100 potential biomarkers for detecting oral cancer have been observed, with IL-8 and IL-1β playing crucial roles in cell replication, angiogenesis, cell adhesion, and tumor development. Several studies have detected IL-1β and IL-8 in the serum and saliva (IL-1β 0.5–12 pg/mL; IL-8 < 62 pg/mL) of patients with oral squamous cell carcinoma (OSCC), highlighting their significance as biomarkers for oral cancer [18,19]. Studies have shown their presence in the serum and saliva of patients with OSCC, underscoring their significance as biomarkers [20]. 

Given the prevalence of oral cancer risk habits in communities like betel nut chewing, alcohol consumption, and smoking, it is hypothesized that these practices may influence salivary profiles, impacting oral health. However, research on the salivary profile within these communities, especially in Flores, East Nusa Tenggara, where such cultural practices are prominent, remains scarce. Hence, this study aimed to assess the salivary profile of individuals with and without oral cancer risk habits in this context.

## 2. Materials and Methods

This observational cross-sectional study involved 77 subjects, comprising 49 individuals with identified risk habits and 28 without such habits, conducted on Flores Island, East Nusa Tenggara province. The determination of the sample size was based on parameters derived from previous studies, specifically an effect size of 1.72 as reported by Amtha et al. (2015), with a study power of 95% and a significance level (α) set at 0.05, calculated using G*Power 3.1.9.7 software [5].

Inclusion criteria encompassed individuals with a history of betel nut chewing, alcohol consumption, and/or smoking for a minimum duration of 1 year. Conversely, exclusion criteria were applied to subjects currently taking antibiotics, antihistamines, or long-term steroids, undergoing cancer treatment, or presenting with cancerous lesions in other organs. All participants provided informed consent before data collection commenced. Ethical approval for the study was obtained from the Ethics Commission of the Faculty of Dentistry, University of Trisakti, under reference number 022/S3/KEPK/FKG/7/2022.

### 2.1. Collection of Unstimulated Salivary Samples

Saliva collection procedures were conducted in the morning, with subjects instructed to refrain from eating, drinking, smoking, or chewing betel quid for a period of 60 min prior to collection. The saliva collection method employed was the spitting technique, where participants were instructed to expectorate saliva into a falcon tube for a duration of 10 min. 

### 2.2. Assessment of the Color, Turbidity, Viscosity, and pH of Saliva

The assessment of color and turbidity was conducted through visual inspection by two calibrated oral medicine specialists. Salivary acidity levels were measured using a digital pH meter (Mediatech pH Meter Digital Automatic Calibration P-2Z), which was calibrated beforehand using 6.86 and 4.00 buffers.

### 2.3. Examination of IL-1β and IL-8 Levels Using an Enzyme-Linked Immunosorbent Assay (ELISA)

Salivary protein separation was conducted using the Genezol Kit (Geneaid, Cat# GZR200). The process involved sample homogenization, followed by the separation of RNA, DNA, and protein in saliva samples. Protein extraction, deposition, washing, and resuspension were then performed to obtain the protein fraction.

Subsequently, the levels of IL-1β and IL-8 (expressed in pg/mL) in saliva were determined utilizing the ELISA method. For this purpose, primers and technical materials specific to IL-1β and IL-8 were utilized (Bioassay Technology Laboratories, Cat# E0143HU, E0089HU). Sample testing was conducted in duplicate to ensure accuracy and reliability.

The optical density values in each well were measured using a microplate reader set to a wavelength of 450 nm. This facilitated the quantification of IL-1β and IL-8 levels in the saliva samples, providing valuable insights into their concentrations.

### 2.4. Data Analysis

Bivariate analysis was conducted using the Chi-square test for non-parametric data and the *t*-test for normally distributed data. A significance level of *p* < 0.05 was deemed statistically significant in determining differences between groups. Odds ratios were calculated to assess the significance of cytokine levels between the groups, providing further insight into the associations observed. Additionally, Kaplan–Meier analysis was used to determine the interleukin level.

## 3. Results

### 3.1. Subject Characteristics

Table 1 shows significant differences in risk factors based on gender (*p* = 0.007). Both males (93.33%) and females (56.4%) exhibited a higher prevalence of risk factors compared to those without such factors. Males primarily presented with smoking and alcohol consumption habits (18.37%), while females predominantly engaged in drinking alcohol and betel quid chewing (38.78%). Additionally, there was a notable disparity in risk factors across age groups (*p* = 0.035), with individuals over 35 demonstrating a higher propensity for risk factors compared to those under 35. Among the 77 study participants, no significant differences were observed in salivary viscosity (*p* = 0.459), color (*p* = 0.476), turbidity (*p* = 1.000), and salivary flow rate (*p* = 0.235) between subjects with and without risk factors.

In this study, the simultaneous consumption of alcohol and betel quid accounts for 38.77% of the population. Additionally, betel quid chewing alone is prevalent (24.48%) in the population. Filter cigarettes are the predominant type of tobacco product used in the region. The preferred alcoholic beverage among community members is “moke putih or tua bhara”. The common composition of betel quid typically includes a mixture of inflorescence betel leaves, unripe areca nut, and lime.

### 3.2. pH, Salivary Flow Rates, and Interleukin Level between Risk Factor Groups

Figure 1 showed the disparities among community cohorts, categorized by the presence or absence of risk factors for oral cancer. Notably, a substantial difference was found in the salivary IL-1β profile between these cohorts, where individuals with identified risk factors exhibit markedly elevated IL-1β levels. Subsequent examination discloses noteworthy variations in IL-1β concentrations between smokers and non-smokers, as smokers demonstrate notably augmented IL-1β levels compared to their non-smoking counterparts. Similarly, a conspicuous distinction emerges in IL-1β concentrations between alcohol consumers and non-consumers, as the former exhibit higher IL-1β levels. Conversely, IL-8 levels do not manifest significant disparities across the cohorts.

Figure 2 describes that IL-1β levels may increase with prolonged use of tobacco, alcohol, and betel quid. Kaplan–Meier analysis showed that the mean estimated time for elevated IL-1β levels was 27.90 months (22.47–33.34). Notably, alcohol consumption appears to accelerate the increase in IL-1β levels more rapidly compared to other oral cancer risk factors.

### 3.3. Correlation between Duration of Risk Factor Habit and Salivary Viscosity

Table 2 shows the absence of a correlation between salivary viscosity and the duration of exposure to risk factors, including smoking, alcohol consumption, or betel nut chewing. 

### 3.4. Comparison of Duration of Risk Factor Habit and Salivary Flow Rate

Table 2 shows that betel nut chewing has been the longest-standing risk factor among the study population, with an average duration of 25.79 years. However, the duration of habits such as smoking, alcohol consumption, and betel nut chewing did not significantly impact the salivary flow rate (*p* > 0.05) (Figure 3). In Figure 4, subjects with a habit of betel nut chewing demonstrate the highest salivary flow rate, followed by those who combine alcohol consumption with betel nut chewing.

## 4. Discussion

Based on gender, the proportion of individuals with risk factors exceeded those without risk factors, 63.64% and 36.36%, respectively (Table 1). The most prevalent risk factor among males is the combined use of alcohol and smoking, consistent with data from the Indonesian Health National Survey indicating a higher prevalence of male smokers (47.3%) compared to females (1.2%) among individuals aged ≥10 years. Conversely, females predominantly engage in the combined consumption of alcohol and betel quid. According to a study by Amtha et al., locals perceive chewing betel leaves as invigorating and fatigue-reducing [5]. Both genders in the Bajawa region exhibit a significant inclination towards alcohol consumption, often attributed to the cold climate, which prompts locals to use alcohol for warmth [5]. Additionally, locals are known for producing homemade alcohol, referred to as “moke”, commonly consumed during traditional ceremonies.

This study showed that individuals aged over 35 exhibit a higher prevalence of risk factors compared to those under 35 (Table 1). In a result consistent with the research conducted by Amtha R et al., a significant proportion of betel quid chewers and alcohol drinkers are elderly individuals [5]. Moreover, data from the Global Adult Tobacco Survey (GATS) 2021 indicates that smoking in the Indonesian population is most prevalent among individuals aged 25 to 44.

In this study, no significant difference was observed in the viscosity of saliva between participants with and without oral cancer risk factors (Figure 1 and Table 2). This lack of disparity can be attributed to the distinct effects of smoking, alcohol consumption, and betel quid chewing on salivary viscosity. Chronic alcohol consumption, as noted by Bronislaw L, leads to increased TNF expression and acinar cell apoptosis, potentially diminishing salivary function and production, consequently elevating saliva viscosity [21]. Studies by Nigar et al. and Petrušić et al. have indicated that smokers typically exhibit thicker saliva compared to non-smokers, potentially as a compensatory mechanism by salivary glands against the continuous deposition of toxins from cigarette compounds [22,23]. Conversely, research by Reddy et al. suggests that prolonged betel quid chewing may lead to a more watery salivation, possibly due to lower levels of potassium and amylase in the saliva of betel nut chewers [24]. In individuals with risk factors, there is no correlation observed between the duration of smoking (*p* = 0.22), alcohol consumption (*p* = 0.151), betel quid chewing (*p* = 0.526), and salivary viscosity (Figure 3). Notably, betel quid chewing poses the second-highest risk factor, resulting in a tendency for saliva viscosity to resemble that of the majority of patients without risk factors.

According to Anand et al., the habit of chewing betel quid often results in the production of blood-red saliva, with the stain becoming ingrained in the teeth, gums, and oral mucosa over time [25]. In the present study, no significant difference in saliva color was observed between patients with risk factors for oral cancer and those without (*p* = 0.476) (Table 1). This lack of distinction may be attributed to the study participants, the majority of whom exhibited high-risk behaviors such as alcohol consumption and betel quid chewing. The presence of high levels of ethanol in saliva due to alcohol consumption can have a solubilizing effect, leading to oral mucosa dryness. Consequently, the color of betel quid constituents may be dissolved, resulting in a lack of discernible differences between patients with and without risk factors.

Areca nut, a key component of betel quid, contains tannin, as identified by Horne et al. [26]. High salivary protein levels, induced by areca nut consumption, can lead to turbidity in saliva. Furthermore, the interaction between tannins and proteins forms a precipitate, resulting in astringency and haze development [26,27]. The study reported both patients with and without risk factors demonstrated a relatively high salivary flow rate, potentially increasing saliva’s protein content and causing cloudiness. Subjects with risk factors also exhibited cloudy saliva, possibly due to lime present in the betel nut. Conversely, in the group without risk factors, turbidity may have been influenced by the limited saliva collection time of 10 min, during which participants typically expectorate saliva, leading to subconscious mucoid engagement and increased salivary turbidity.

In this study, no significant difference was found in the salivary flow rates between individuals with and without oral cancer risk factors (*p* = 0.235) (Table 1 and Figure 4). Notably, subjects with a singular habit of betel quid chewing exhibited the highest salivary flow rate, consistent with findings by Rooban et al., who reported that betel quid chewers had greater salivary flow rates compared to non-chewers [28]. This phenomenon is attributed to the enlargement of salivary glands resulting from persistent betel quid chewing, leading to an elevated salivary flow rate [28]. Contrary to the above, chronic alcoholics are prone to experiencing dry mouth, as suggested by Innenaga et al. The accumulation of ethanol and acetaldehyde can induce apoptosis and cell death in acinar cells, while alcohol intake may also lead to fat accumulation in salivary glands, acinar cell swelling, atrophy, and alterations in salivary flow rate, resulting in decreased salivary production [29]. Additionally, the presence of nicotine, as observed by Nigar et al. and Petrusic et al., may initially boost salivary flow rate but is followed by a subsequent decrease over time [2,22]. However, in the current study, no significant difference in salivary flow rate was noted. This discrepancy may be explained by the interaction between alcohol consumption and betel quid chewing in this demographic. While alcohol consumption tends to result in a drier oral cavity, betel quid chewing is associated with a higher salivary flow rate. When compared to individuals without risk factors, who typically exhibit normal salivary flow rates, there was no notable difference in salivary flow rate among this cohort. Furthermore, the duration of risk factor behavior (smoking, alcohol consumption, or betel quid chewing) did not affect salivary flow rate in this study. However, it is important to acknowledge that various factors such as stress, medication use, and salivary gland diseases may influence salivary production, potentially leading to xerostomia. These findings diverge slightly from earlier research, suggesting the presence of additional factors that may impact salivary flow rate [30]. 

According to Lin et al., Taiwanese natives commonly engage in the practice of betel quid usage, often accompanied by drinking and smoking [31]. Similarly, research by Rae et al. in India, Sri Lanka, and Pakistan reveals frequent engagement in bidi smoking, alcohol consumption, and chewing tobacco containing tobacco [32]. This suggests that populations across diverse countries face a multifactorial risk of oral cancer. Consistent with this notion, the population in Bajawa exhibits a variety of risk factor combinations, particularly involving the simultaneous consumption of alcohol and betel quid, compared to a single habit such as betel nut chewing alone. The synergistic effect of multiple hazardous behaviors, including betel quid chewing, smoking, and alcohol consumption, heightens the risk of Oral Potentially Malignant Disorders (OPMDs) and oral cancer [33]. In contrast to the findings of Amtha et al. in the Tanjung Pandang population, where individuals tended to have a single smoking habit, the average population displayed normal variant lesions and trauma lesions rather than OPMD lesions. This discrepancy may be attributed to the presence of only one risk factor for oral cancer in this group, which may be insufficient to cause OPMD lesions or oral cancer [34].

Comprising betel leaf, areca nut, and lime, betel quid represents the most prevalent form of chewing tobacco among the population of Flores, particularly in Bajawa. These findings align with qualitative research conducted concurrently by other scholars, indicating that chewing betel quid is widespread among the majority of the Bajawa population [5]. In western Indonesia, the combination typically includes betel leaf, gambier, and wet lime, whereas in eastern Indonesia, betel fruit, unripe areca nut, and dried lime are more commonly utilized [35]. According to the research findings of Sari et al., the levels of polyphenols and arecoline in areca nut seeds are directly correlated with their ripeness [36]. Young areca nuts are deemed more carcinogenic due to their higher concentrations of polyphenols and arecoline compared to older nuts [36]. Variations in the composition and usage practices of betel quid can lead to differing risks of oral cancer among individuals [35]. 

In this study, no significant difference was observed in the acidity of saliva between individuals with and without oral cancer risk factors (Figure 1). This finding contrasts with previous research suggesting an increase in saliva pH among betel quid chewers. Additionally, Priyanka et al.’s study reported that although not statistically significant, the alcohol-drinking group exhibited lower salivary pH compared to the control group [37]. According to Singh et al., long-term smoking can lead to decreased salivary pH and flow rate [38]. The lack of a significant variation in salivary pH in our study may be attributed to similar saliva production rates between the betel nut and control groups. Moreover, the quantity of tobacco, lime, and other constituents consumed could potentially influence salivary pH. Lime contains a substantial concentration of alkali, which may impact the salivary buffer system’s pH [28]. Furthermore, the impact of alcohol on the oral cavity depends on factors such as the beverage’s composition, qualities, frequency, and quantity of consumption [37]. Additionally, salivary flow rate, which did not significantly differ between groups with and without risk factors in our study, may also affect saliva pH. Low salivary flow rates result in reduced bicarbonate concentration, leading to a decrease in salivary pH, and vice versa [38]. 

Interleukin-1 beta (IL-1β) has emerged as a pivotal player in the complex landscape of oral malignancies within the oral tumor microenvironment. The study made by Lee et al. regarding IL-1β as a mediator triggered by chronic inflammation, along with findings from previous studies, suggests that risk factors such as betel quid chewing, alcohol consumption, and smoking act as stimuli for inflammation, resulting in increased cytokine expression [23]. The Kaplan–Meier analysis depicted in Figure 2 shows the cumulative increase in IL-1β levels over time among individuals with different habits such as smoking, alcohol consumption, and betel quid chewing (BQC), both individually and in combination. The data indicate that alcohol consumption leads to a more rapid increase in IL-1β levels compared to smoking and BQC alone, suggesting a more immediate inflammatory response. When these habits are combined, the effect on IL-1β levels is even more pronounced, particularly in the group combining smoking, alcohol, and BQC, which shows the highest levels over time. This synergistic effect underscores the compounded risk of engaging in multiple harmful habits, contributing to elevated inflammatory markers. This condition is supported by the results of the study conducted in the Bajawa population, which revealed significantly higher levels of IL-1β in saliva among smokers compared to non-smokers. The heat generated by cigarette smoke and the interaction of nicotine with nitrites, leading to the formation of specific nitrosamines like 4-(methylnitrosamino)-1-(3-pyridyl)-1-butanone and 4-(methylnitrosamine)-1-(3-pyridil)-1-butanol, are known to trigger an inflammatory response. Prolonged exposure to cigarette smoke can induce a chronic inflammatory process, involving the recruitment of inflammatory cells, particularly macrophages. These macrophages play a crucial role in oral cancer development by releasing pro-inflammatory cytokines such as IL-1β and IL-8. Additionally, they contribute to the elevation of reactive oxygen species (ROS). Oxidative stress, resulting from an increase in ROS levels, leads to DNA damage and genomic instability, which are recognized mechanisms in cancer development. Therefore, the observed elevation in IL-1β levels in smokers could be indicative of the underlying inflammatory processes contributing to the pathogenesis of oral cancer [39]. 

The findings of this population-based study revealed a noteworthy disparity in IL-1β levels between individuals who consumed alcohol and those who did not, with higher IL-1β levels observed among alcohol consumers (Figure 1). Chronic alcohol consumption has been implicated in damaging oral mucosal tissue and triggering inflammation, which can subsequently induce IL-1β production. Key processes contributing to this phenomenon include oxidative stress and the breakdown of alcohol metabolites. Chronic alcohol intake leads to the production of reactive oxygen species (ROS) and reactive nitrogen species (RNS) during the conversion of ethanol into acetaldehyde. When the generation of ROS and RNS surpasses the body’s antioxidant capacity, oxidative stress ensues. These free radicals are highly reactive and perceived as foreign agents by the immune system, prompting an enhanced secretion of cytokines and chemokines, including IL-1β [40]. Moreover, alcohol consumption interferes with the synthesis of collagen and other essential proteins crucial for tissue repair and regeneration. Consequently, the impaired production of these proteins impedes the healing process of damaged oral mucosa, exacerbating persistent inflammation [41]. These mechanisms underscore the association between alcohol consumption and elevated IL-1β levels observed in this study, highlighting the detrimental effects of chronic alcohol use on oral health.

The research findings indicate a significant disparity in IL-1β levels between patients with and without risk factors (Figure 1 and Figure 2). The observed elevation in IL-1β levels among individuals with risk factors aligns with the chronic inflammatory responses elicited by these factors. Previous studies have established a correlation between increased IL-1β levels and pathological changes conducive to the development of oral potentially malignant disorders (OPMDs) and cancer. Furthermore, it is plausible that genetic predispositions within this population contribute to their heightened susceptibility to cancer development [39]. This suggests a multifactorial etiology, wherein both environmental exposures and inherent genetic susceptibilities synergistically influence disease pathogenesis. This cytokine not only promotes tumor growth and invasion by stimulating angiogenesis and enhancing matrix metalloproteinases (MMPs) expression but also fosters an immunosuppressive milieu that aids in tumor evasion from immune surveillance. These findings underscore the intricate interplay between genetic factors, environmental exposures, and inflammatory processes in the initiation and progression of oral cancer.

According to studies conducted by Mio et al., the presence of acrolein and acetaldehyde in cigarette smoke can stimulate bronchial epithelial cells to produce higher levels of IL-8 [42]. Similarly, research by Huang et al. indicates that ethanol exposure can elevate plasma endotoxin levels and trigger the production of TNF-α and IL-1 [43]. These pro-inflammatory cytokines contribute to the augmentation of local and systemic IL-8 production in patients with alcoholic liver disease (ALD) [43]. In contrast to the marked increase observed in IL-1β levels, the rise in IL-8 levels appears to be less pronounced in this study. However, elucidating a direct correlation is complex. Sahibzada et al. demonstrated that various factors beyond exposure to risk factors can influence IL-8 levels, including lifestyle choices, geographical disparities, ethnic variations, genetic predispositions, and individual habits [44]. Moreover, findings from Vychaktami et al. suggest that certain dietary components commonly found in fruits and vegetables, such as curcumin, aloe vera, quercetin, and lycopene, possess anti-inflammatory properties and can inhibit the production of cytokines implicated in inflammation, including IL-1, IL-2, IL-6, IL-8, and IL-12 [45]. Consequently, it is plausible that the consumption of these dietary components may mitigate the elevation of IL-8 levels in individuals with risk factors [46]. This multifaceted interplay underscores the intricate mechanisms underlying cytokine regulation and the potential mitigating effects of dietary interventions on inflammation-associated pathways.

This study possesses several limitations that warrant acknowledgment. Firstly, the study did not account for the periodontal health status of participants, which may influence cytokine release in saliva. Secondly, the absence of information regarding the type of areca nut (unripe or overripe) utilized for betel nut chewing is pertinent, as it affects the alkaloid content in areca nut, potentially contributing to its carcinogenic properties. Additionally, the restricted saliva collection duration of 10 min may have led participants to inadvertently include mucoid substances in their samples, thereby impacting the assessment of salivary turbidity. Despite these limitations, the study’s notable strength lies in being the first investigation of saliva profiles among individuals with oral cancer risk factors in East Indonesia, a region characterized by a high prevalence of such risk factors. As a recommendation for future research, it may be beneficial to include an evaluation of salivary antioxidants in relation to oral cancer susceptibility, thereby providing further insights into potential biomarkers for early detection and prevention strategies.

## 5. Conclusions

The present study uncovered differences in salivary IL-1β concentrations between individuals with and without habits linked to oral cancer risk. Nonetheless, no disparities were detected in salivary viscosity, color, turbidity, flow rate, acidity, or IL-8 levels. These findings suggest that alterations in salivary profiles may hold promise as early indicators and novel diagnostic tools for identifying abnormalities within the oral cavity.

## Figures and Tables

**Figure 1 biomedicines-12-01748-f001:**
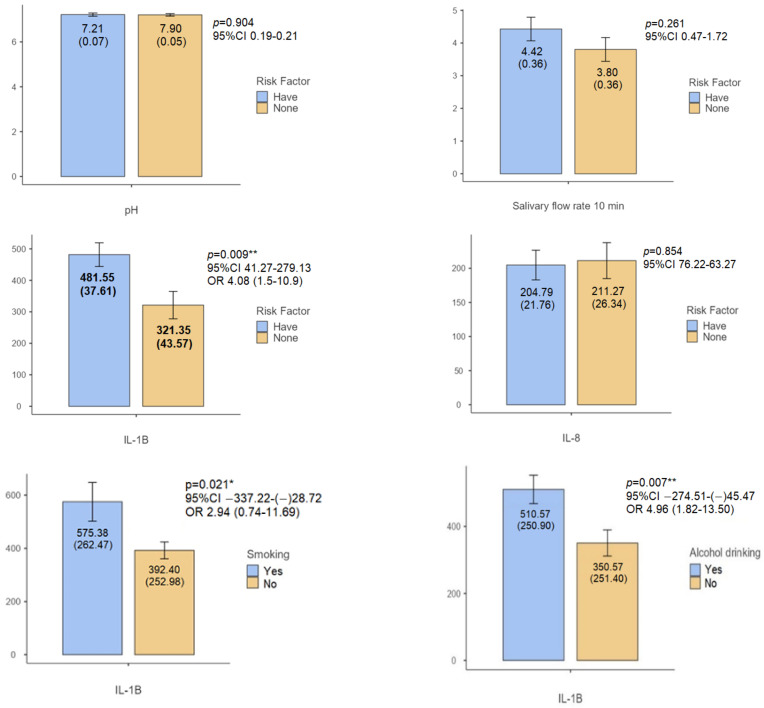
Bar chart showing the mean and standard deviation of pH, salivary flow rate, IL-1β, IL-8 among the studied groups. ** *p* < 0.01 * *p* < 0.05.

**Figure 2 biomedicines-12-01748-f002:**
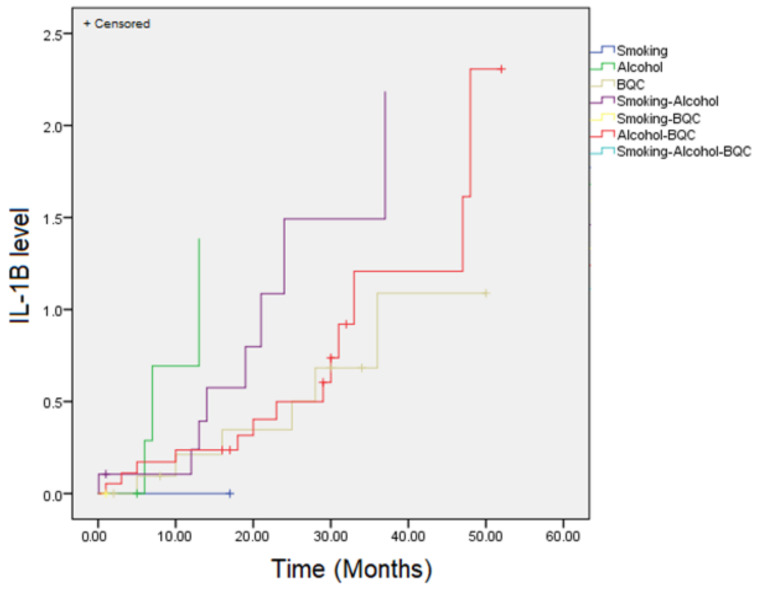
Kaplan–Meier analysis of time to IL-1β level among the groups.

**Figure 3 biomedicines-12-01748-f003:**
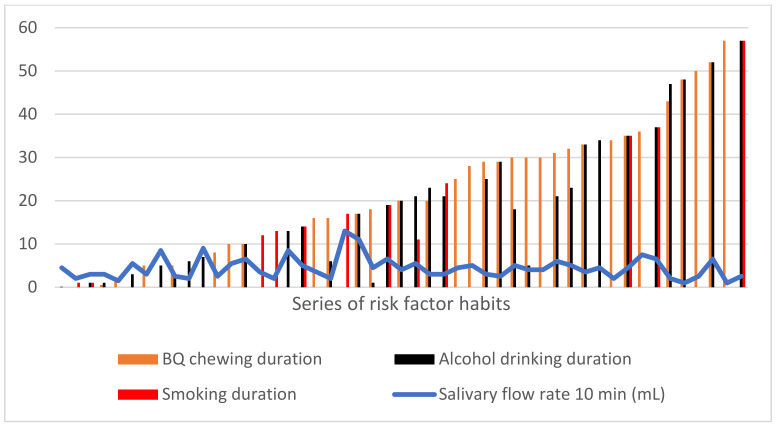
Series of risk factor habits based on the duration and salivary flow rate 10 min. (BQC, betel quid chewing).

**Figure 4 biomedicines-12-01748-f004:**
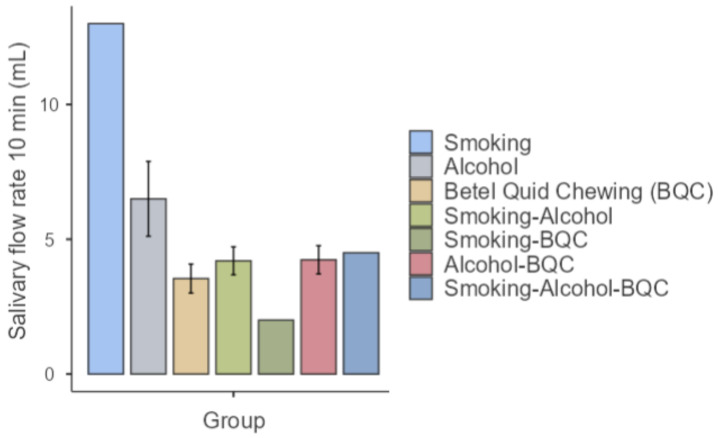
The salivary flow rates over a 10 min period vary across different risk factor habit groups as follows: smoking: 13.0 mL; alcohol: 6.50 (3.10) mL; betel quid chewing (BQC): 3.54 (1.86) mL; smoking–alcohol: 4.20 (1.64) mL; smoking–BQC: 2 mL; alcohol–BQC: 4.24 (2.28) mL; smoking–alcohol–BQC: 4.50 mL.

**Table 1 biomedicines-12-01748-t001:** Population characteristics.

Variable	Risk Factor		Non-Risk Factors		Total	*p*-Values
	n	%	n	%	n	
Gender						0.008 **
Male	14	93.33	1	6.67	15	
Female	35	56.45	27	43.55	62	
Age (years)						0.035 *
<35.3	12	46.15	14	53.85	26	
>35.3–42.7	16	64	9	36	25	
>42.7	21	80.77	5	19.23	26	
Salivary viscosity						0.459
Dilute	30	60	20	40	50	
Thick	19	70.37	8	29.63	27	
Color						0.476
Colorless	29	60.42	19	39.58	48	
Colored	20	68.97	9	31.03	29	
Turbidity						1.000
Clear	8	61.54	5	38.46	13	
Cloudy	41	64.06	23	35.94	64	
Salivary flow rate (mL/10 min)						0.235
<3.2	20	55.55	16	44.44	36	
>3.2	29	70.73	12	29.27	41	
Risk factor habit						
Smoking (S)	1	2.04	-	-	1	
Alcohol drinking (A)	5	10.2	-	-	5	
BQ Chewing (BQ)	12	24.48	-	-	12	
S + A	10	20.40	-	-	10	
S + BQ	1	2.04	-	-	1	
A + BQ	19	38.77	-	-	19	
S + A + BQ	1	2.04	-	-	1	
BQ Chewing composition						
Betel leaf, areca nut, lime	16	100	-	-	16	
Betel leaf, areca nut, lime, tobacco	1	100	-	-	1	
Betel fruit, areca nut, lime	2	100	-	-	2	
Betel fruit, betel leaf, areca nut, lime	13	100	-	-	13	
Betel fruit, betel leaf, areca nut, lime, gambier, tobacco	1	100	-	-	1	

S, smoking; A, alcohol drinking; BQ, betel quid; * *p* < 0.01; ** *p* < 0.05.

**Table 2 biomedicines-12-01748-t002:** Salivary viscosity and duration of risk factor habit.

Variable	Duration (Year)					
	Smoking		Alcohol Drinking		Betel Quid Chewing	
	Mean (SD)	*p*	Mean (SD)	*p*	Mean (SD)	*p*
Salivary viscosity		0.22		0.151		0.526
Dilute	22.94 (11.91)		16.60 (12.92)		17.42 (11.25)	
Thick	29.73 (18.28)		24.84 (19.30)		23.80 (22.26)	
Duration habit	20.08 (16.12)		19.85 (15.98)		25.79 (15.02)	

## Data Availability

The data that support the findings of this study are available on request from the corresponding author, I.G. The data are not publicly available due to information that could compromise the privacy of research participants.

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
