# Peer review of "Salivary Profile Analysis Based on Oral Cancer Risk Habits: An Observational Cross-Sectional Study"

_biomedicines, 2024, doi:10.3390/biomedicines12081748_

Round 1

Reviewer 1 Report

Comments and Suggestions for Authors

Overall, I found the article well-structured and with a clear methodology. The results are discussed in detail, and the findings match the presentations in the tables and figures.

However, several minor English and grammatical errors should be corrected for better clarity and readability.

For instance,

Abstract:

 "Purpose: to compare" should be "Purpose: To compare".

"turbidity, and levels of IL-1β and IL-8." should be "turbidity, and the levels of IL-1β and IL-8."

"various parameters were measured" should be "Various parameters were measured".

"Data were analyzed using Chi-square and t- tests." should be "Data were analyzed using Chi-square and t-tests."

"No significant differences were found between the groups in terms of" can be improved to "No significant differences were observed between the groups in terms of."

Introduction:

"WHO estimates that the majority of oral and oropharyngeal cancer cases globally, around 58%, occur in South Asia and Southeast Asia, including Indonesia." This can be simplified for clarity: "WHO estimates that around 58% of global oral and oropharyngeal cancer cases occur in South and Southeast Asia, including Indonesia."

The citation format should be consistent throughout the document. For example, "[2];[3]" should be "[2,3]".

"and non-routine smokers as high as 7.3%, as well as the second highest province with a proportion" should be "and non-routine smokers at 7.3%. It is also the second highest province with a proportion".

Results:

"Table 1 showed significant differences" should be "Table 1 shows significant differences".

"In this study, a significant risk factor for oral cancer is the simultaneous consumption" should be "In this study, the simultaneous consumption."

"Notably, the habit of alcohol consumption appears" should be "Notably, alcohol consumption appears."

Comments on the Quality of English Language

Needs thorough language edits. 

Reviewer 2 Report

Comments and Suggestions for Authors

The aim of the present investigation was to compare the salivary profiles of individuals with and without risk factors for oral cancer.

GENERAL COMMENTS

The article is in-line with the journal topic, but flaws should be improved.  The investigation is interesting, and the present paper is recommended for publication to the present journal after major revision.

Title: The title should indicate the type of study that has been conducted: (f.e.: Salivary profile analysis based on oral cancer risk habits: an observational cross-sectional study)

Introduction

1.      The authors should support the biological basis of the biomarker chosen.

2.      Il-1beta and IL-8 seems to be pro-inflammatory aspecific markers involved in carcinogenesis but also a wide quantity of oral disease. The interleukins detected in oral saliva could easily misinterpreted (f.e. periodontitis, gingivitis….). The physiological level of salivary Il-1beta and IL-8 should be declared.

3.      The authors considered color, turbidity, viscosity, and pH of saliva as potential predictors for oral carcinogenesis risk. If applicable, the authors should support the biological bases with references.

Materials and methods

1.      The inclusion and exclusion criteria section is missed.

2.      Did you considered also subjects with oral cancer?

3.      “Saliva collection procedures were conducted in the morning, with subjects instructed to refrain from eating, drinking, smoking, or chewing betel quid for a period of 60 minutes prior to collection”.  In my opinion, also oral rinses with antiseptics and teeth brushing could modify the salivary properties.

4.      The present study design did not take in account any quantitative predictor for smoking, betel fruit, betel leaf, areca nut, lime consumption. What was the inferior limit of consumption/die for each predictor?

5.      A repository for raw metadata access should be added to the present study as supplemental material.

Results

The result section is weak. The present investigation did not consider other independent risk factor (f.e. familiarity). In my opinion, the validation process should take in account a stronger study design for this purpose (f.e. RCT).

The authors did not take in account the combination of risk factor in Il-1beta and IL-8. It could be interesting to verify the power effect and the synergy of the predictors considered.

Discussion

The authors should discuss the oral cancer epidemiology, physiopathology, clinical course, TNM and characteristics. The risk factors should be discussed in a more accurate way. In addition, several potential salivary biomarkers (including mRNA detection) have been purposed for this scope as in-chair test for primary prevention. The future perspective of this application is certainly interesting but the critical limit of the present study design is the model validation. The Il-1beta and IL-8 correlation with oral cancer is weak and too aspecific for a cross-sectional study design.. In contrary, the study model should include a test group (f.e. patients affected by oral cancer) and a control group.

Reviewer 3 Report

Comments and Suggestions for Authors

This is an interesting study on salivary profile analysis of oral cancer risk habits in Indonesian patients.

The article is well written, clear to read, and the results are well presented.

The Tables are easy to read, only Figure 2 could be slightly enlarged.

Here are my comments:

The introduction could be a little more extensive on saliva (role, properties, etc.).

In the materials and methods section, it would be interesting not only to provide the duration of the habits of the patients, but also the daily amount.

It would have been interesting to have a control group of patients with precancerous lesions or squamous cell carcinoma.

In the discussion, the role of IL1β in the process of malignant transformation should be further discussed.

Finally may be 49 patients is a limited number of tested subjects

Round 2

Reviewer 2 Report

Comments and Suggestions for Authors

The paper has been improved